# Overexpression of ERAP2N in Human Trophoblast Cells Promotes Cell Death

**DOI:** 10.3390/ijms22168585

**Published:** 2021-08-10

**Authors:** Kristen Lospinoso, Mikhail Dozmorov, Nadine El Fawal, Rhea Raghu, Wook-Jin Chae, Eun D. Lee

**Affiliations:** 1Department of Microbiology and Immunology, School of Medicine, Massey Cancer Center, Virginia Commonwealth University, Richmond, VA 23298, USA; lospinosokr@mymail.vcu.edu (K.L.); elfawalna@vcu.edu (N.E.F.); Wook-Jin.Chae@vcuhealth.org (W.-J.C.); 2Department of Biostatics and Pathology, Massey Cancer Center, Virginia Commonwealth University, Richmond, VA 23298, USA; mikhail.dozmorov@vcuhealth.org; 3Tenafly H.S., Tenafly, NJ 07670, USA; 23rraghu@tenafly.k12.nj.us

**Keywords:** ERAP2, trophoblast cells, pregnancy, pre-eclampsia, RNA sequencing, differentially expressed genes

## Abstract

The genes involved in implantation and placentation are tightly regulated to ensure a healthy pregnancy. The endoplasmic reticulum aminopeptidase 2 (ERAP2) gene is associated with preeclampsia (PE). Our studies have determined that an isoform of ERAP2-arginine (N), expressed in trophoblast cells (TC), significantly activates immune cells, and ERAP2N-expressing TCs are preferentially killed by both cytotoxic T lymphocytes (CTLs) and Natural Killer cells (NKCs). To understand the cause of this phenomenon, we surveyed differentially expressed genes (DEGs) between ERAP2N expressing and non-expressing TCs. Our RNAseq data revealed 581 total DEGs between the two groups. 289 genes were up-regulated, and 292 genes were down-regulated. Interestingly, most of the down-regulated genes of significance were pro-survival genes that play a crucial role in cell survival (LDHA, EGLN1, HLA-C, ITGB5, WNT7A, FN1). However, the down-regulation of these genes in ERAP2N-expressing TCs translates into a propensity for cell death. The Kyoto Encyclopedia of Genes and Genomes (KEGG) analysis showed that 64 DEGs were significantly enriched in nine pathways, including “Protein processing in endoplasmic reticulum” and “Antigen processing and presentation”, suggesting that the genes may be associated with peptide processes involved in immune recognition during the reproductive cycle.

## 1. Introduction

The placenta is the largest organ in the fetus; it begins developing with the implantation of the blastocyst into the uterine wall by the end of the first week after fertilization in humans [1,2]. After the 16-cell stage, the morula begins differentiating into the two major cell types of a blastocyst: an internal cell mass that becomes the embryo and an outer layer of trophoblastic cells (TCs) that mostly become the placenta [1,2]. Through a tightly regulated process, the trophoblasts differentiate into two major cell types: an inner layer of cytotrophoblasts and an outer layer of syncytiotrophoblasts that line the placental villi that invade the endometrium [3]. As fetal trophoblasts invade maternal tissue, immunologic adjustments occur in the decidua to ensure the establishment of an immune environment favorable to fetal growth [4]. In a normal pregnancy, myeloid dendritic cells (DCs) with immature phenotypes secrete anti-inflammatory cytokines and promote fetal tolerance, along with antigen-specific regulatory T (Treg) cells [5]. In addition to their inflammatory function to protect against pathogens, Natural Killer (NK) cells in the decidua reinforce uterine vasculature development, thereby supporting fetal growth demands [6]. The abnormal expression or function of any of these immune modulators is implicated in several maternal and fetal pathologies, including pre-eclampsia (PE) [5].

The responses of immune cells after implantation can be modulated via endoplasmic reticulum aminopeptidase (ERAP)1 and ERAP2 expression in TCs. These are enzymes located in the endoplasmic reticulum whose primary function is to trim peptide antigens for presentation on MHC Class I molecules [6,7]. In addition to their antigen processing function, the ERAP1 and ERAP2 proteins have been associated with the development of autoimmune disorders, eclampsia, and PE, respectively [7,8,9].

Our group and others have reported that the ERAP2 gene is genetically linked to PE in multiple distinct patient populations; the literature highlights Norwegian, Australian/ New Zealand, and African American patients, in particular [8,9]. During a continued surveillance for genetic links to PE, our study discovered a Chilean population that did not show the genetic link of ERAP2 to PE. The haplotype of ERAP2 found in this specific Chilean population is a novel one, and its role in pregnancy and potential for immunogenicity has yet to be explored [10]. PE is not the immediate focus of our study. However, our goal of gathering data on ERAP2’s influences on the intrinsic expression of genes will increase our understanding of its role in immunogenic responses and abnormal placentation in conditions such as PE.

ERAP2 exists in two isoforms based on a single nucleotide polymorphism (SNP), rs2549782. ERAP2N has an asparagine (N) present at the 392 positions. ERAP2K has a lysine residue (K) present at the same position. The polymorphism leads to a conformational change in the catalytic site of the protein [11]. This conformational change results in altered antigen processing and thus may have significant implications for immune modulation. ERAP2N has approximately 165-fold greater activity for hydrophobic amino acids than the ERAP2K protein, potentially increasing the antigen presentation to the host immune system [11,12]. Previous studies have demonstrated that Haplotype A (ERAP2K) and Haplotype B (ERAP2N) are selected for frequencies of 0.44 and 0.56, respectively; however, Haplotype B undergoes differential splicing and nonsense-mediated mRNA decay. As a consequence, Haplotype B is never observed in nature [13]. Notably, there was a complete absence of homozygosity of ERAP2N in the maternal and fetal genetic screening in Chilean populations [10]. This suggests that the expression of ERAP2N is incompatible with life in utero. The explanation for this observation is demonstrated in our previous study. We observed that when ERAP2N TCs are co-cultured with peripheral blood monocytic cells (PBMCs), there is an increased immune activation of the CTL- and NKC-induced apoptosis of these cells in vitro [12,14].

The exact mechanism as to how ERAP2N expression leads to increased apoptosis has yet to be outlined in the literature. Before investigating further, the possible heightened immune response elicited by ERAP2N, we opted first to characterize the RNA-level changes observed with or without ERAP2N-expressing TCs. While the ultimate aim of our research is to apply these findings to live human subjects, culture cells (JEG-3) were utilized in this experiment rather than human trophoblast cells from terminated specimens. This was due to the logistical and ethical difficulties associated with obtaining terminated specimens of Chilean populations that would express ERAP2N, as well as the fact that our experiment is a preliminary attempt to explore the differences, and thus cultured cells are adequate for reaching our immediate goals. It is our hope that, in the future, the application of similar experimental protocols may be performed with human tissue to ensure that the results are compatible with our initial reports in this study. To accomplish this, we performed an unbiased mRNA sequencing analysis of ERAP2N-positive TCs compared to those without ERAP2N. Because we have shown previously that ERAP2N-expressing TCs increased immune cell-induced apoptosis, we highlight RNA expression changes in the category of pro-proliferation/survival or pro-apoptosis/death [12,13,14,15]. This study aims to elucidate key gene alterations and whether the gene expression profile may explain the increased apoptotic activity in TCs that express ERAP2N.

## 2. Results

### 2.1. Sample Identification

Total RNA was extracted from three different ERAP2N-positive JEG-3 stable transfectants and three stable transfectants with an empty vector without ERAP2N as a control. The paired-end transcript reads from each of these groups were aligned to the hg38/GRCh38 reference genome using the Spliced Transcripts Alignment to a Reference (STAR) aligner v2.6.1 [16].

### 2.2. An Overview of Differentially Expressed Genes in ERAP2N-Positive vs. ERAP2N-Negative Trophoblast Cells

The volcano plot represents the global overview of genes that were differentially affected by the ERAP2N expression in human TCs compared to the TCs that do not express ERAP2N (Figure 1). The differential expression analysis using the edgeR method using the adjusted p-value cutoff <0.3 identified 581 genes differentially expressed between the two groups [17]. Overall, 289 genes were up-regulated and 292 genes were down-regulated, indicating the balanced shift in gene expression changes induced by ERAP2N overexpression. 

### 2.3. The Heatmap of the Top 50 Most Significantly Differentially Expressed Genes in ERAP2N-Positive vs. ERAP2N-Negative Trophoblast Cells

The heatmap represents the cluster of the top 50 genes, in which the first three rows show the gene expression in ERAP2N-negative TCs, while the three rows on the right display the gene expression in ERAP2N-positive TCs (Figure 2). Interestingly, the top half of the heatmap cluster genes appears to be mostly involved in pro-apoptotic pathways, while the bottom half centers on those involved in cell survival. In ERAP2N-positive TCs, pro-apoptotic cells are up-regulated compared to ERAP2N-negative control cells, in which these same genes are down-regulated. By contrast, pathways that support cell survival are down-regulated in ERAP2N-positive TCs but up-regulated in those that are ERAP2N-negative.

### 2.4. Functional Enrichment Analysis of KEGG Pathways

Table 1 lists the major KEGG pathways that were significantly enriched with an adjusted *p*-value < 0.3. The entire list of significant pathways at the adjusted *p*-value > 0.3 is in the Appendix A. The color-coded genes are 12 highly differentially expressed genes (top 50 DEGs) that were also found in these major KEGG pathways which we believe might be significant in determining the cell survival rate with ERAP2N-positive TCs. Specifically, these 12 genes (FN1, LDHA, WNT7A, EGLN1, ITGB5, HLA-C, PPP1R12B, ENO3, CDC42, EGLN2, THBS3, SEC31B) were found to be involved in the following enriched pathways, some genes overlapping in multiple pathways. The rest of top 50 genes divided into cell survival and cell death is listed in the Appendix A.

The list of differentially expressed genes was also analyzed for enrichment in the Kyoto Encyclopedia of Genes and Genomes (KEGG) pathways using ShinyGO (Figure 3). All pathways are indicated through green nodes and are considered “connected” if they share 20% or more genes. The darker the node, the more significantly enriched the gene sets, and the larger the node, the larger the gene sets. Glycolysis/Gluconeogenesis, Protein processing in the endoplasmic reticulum, and the HIF-1 signaling pathway all seem to be the pathways with the most significantly enriched gene sets. Since thicker lines delineate more overlapped genes between the pathways, or nodes, Carbon metabolism, Glycolysis/Gluconeogenesis, and the Biosynthesis of amino acids contain more overlapped genes between the pathways as compared to the other significant pathways. Interestingly, Protein processing in the endoplasmic reticulum shares no overlapped genes with the rest of the pathways.

### 2.5. Differentially Expressed Genes (DEGs) in ERAP2N-Positive Cells Compared to ERAP2N-Negative Cells

The S-curve (Figure 4A) is a plot of genes differentially expressed in ERAP2N-positive TCs compared to ERAP2N-negative empty vector TCs, ranked by log2 fold change. The genes plotted on the blue curve are those that are significantly down-regulated, while those on the red curve are significantly up-regulated in ERAP2N-positive TCs, compared to TCs that do not express ERAP2N. LogFC measures how much the gene expression changes between two different conditions. A negative LogFC indicates that the expression of a particular gene is decreasing (i.e., it is being down-regulated) in ERAP2N-positive cells, relative to TCs that are ERAP2N-negative. In the same respect, a positive LogFC indicates that a gene is up-regulated, or that its expression increases in ERAP2N-positive cells compared to cells that do not express ERAP2N. We chose to highlight genes of interest that are relevant in cell survival or cell death. The majority of the genes that are necessary for placental function and cell survival, such as WNT7A, appear to become down-regulated in ERAP2N-positive cells [18,19,20]. In addition, apoptosis-linked genes such as PPP1R12B and EGLN2 appear to be up-regulated in ERAP2N-positive trophoblasts [21,22].

Next, the expression level of each individual gene between the ERAP2N-positive and cells without ERAP2N is directly compared in Figure 4b using the barplot of 12 differentially expressed genes (DEGs). The expression level is reported as Log2 TPM (TPM= transcript count per million). TPM scales the read count of RNA transcripts of a particular gene to the total read count of the sequencing run, thereby measuring the amount of RNA in a sample. The genes represented by the bar plots are a subset of many genes for which the amount of RNA was significantly different in ERAP2N-positive trophoblasts compared to its control. Similarly, PPP1R12B, ENO3, CDC42, THBS3, EGLN2, and SEC31B are up-regulated with ERAP2N, whereas HLA-C, ITGB5, EGLN1, WNT7A, LDHA, and FN1 are down-regulated.

### 2.6. Top 12 Genes in Cellular Fate Expressed in ERAP2N-Positive Cells

Even though the highlighted genes are divided approximately equally between up-and down-expression levels, Figure 5 clearly depicts how these genes determine the fate of ERAP2N-expressing TCs. According to the literature, out of these 12, eight genes (LDHA, EGLN1, HLA-C, ITGB5, WNT7A, FN1, PPP1R12B, and EGLN2) favored cell death depending on their expression level: LDHA, EGLN1, HLA-C, ITGB5, WNT7A, and FN1 are reduced, whereas PPP1R12B and EGLN2 are increased [19,20,21,22,23,24,25,26,27,28,29,30]. The six genes labeled green (LDHA, EGLN1, HLA-C, ITGB5, WNT7A, FN1) promote cell death due to their main function of maintaining the homeosis that promotes cell survival; however, these genes promote cell death when their expression level is down-regulated in ERAP2N-positive TCs, as indicated with the blue down arrow. The two red genes (PPP1R12A, EGLN2) are pro-cell death when up-regulated, so naturally, in ERAP2N-positive cells, they favor cell death, indicated with the red up arrow [19,20,21,22,23,24,25,26,27,28,29,30]. Interestingly, ERAP2N-positive TCs still have four green genes that promote cell survival (ENO-3, CDC42, THBS3, SEC31B) when their expressions are up-regulated [31,32,33,34]. Nonetheless, the diagram supports our hypothesis that ERAP2N expression in trophoblast cells alters gene expression levels and patterns to favor cell death.

The pathway scheme in Figure 6 illustrates the association between ITGB5 and WNT, which explains how these two genes of significance were down-regulated with the presence of ERAP2N promoting cell death instead of cell proliferation/survival. The diagram shows that ITGB5 associates with FAK to activate the Erk cascades that promote transcription. In addition, β-catenin and Wnt contribute to this process through the β-catenin/Wnt signaling pathway. When Wnt binds to the Frizzled receptor, disheveled I (Dvl) is activated, which, through β-catenin, also leads to the Wnt target gene transcription. Thus, both ITGB5 and Wnt promote the proliferation and differentiation of the cell, leading to a pro-survival cell fate.

## 3. Discussion

Trophoblast cells (TCs) of the placenta invade the maternal uterine wall during implantation and are critical in providing adequate signaling to neighboring maternal immune cells to orchestrate a healthy pregnancy [35]. However, we have recently reported that cytotoxic T-lymphocytes (CTLs) and natural killer (NK) cells are significantly activated and increased, preferentially targeting and killing ERAP2N-expressing TCs, potentially disrupting normal pregnancy [16]. This may explain why Chilean populations did not display any ERAP2N homozygosity [10].

To explore the mechanism of ERAP2N and to gain a deeper understanding of the biology of TCs, we assessed changes that may indicate susceptibility to cell death. We successfully identified the gene expression profiles of TCs with or without ERAP2N using RNA-Seq (Quantification) and analyzed the gene expression differences between the two libraries. Furthermore, the KEGG pathways for the DEGs were analyzed.

In our study, a total of 581 genes were significantly differentially expressed between the two groups. A large number of DEGs are associated with the following pathways with an adjusted *p*-value more significant than 0.3: Glycolysis/Gluconeogenesis, HIF-1 signaling pathway, Fructose and mannose metabolism, Protein processing in the endoplasmic reticulum, Antigen processing and presentation, Central carbon metabolism in cancer, Proteoglycans in cancer, ECM-receptor interaction, and Human papillomavirus infection (Table 1). As represented in Figure 2, all these pathways are highly associated with each other, except for the protein processing in the endoplasmic reticulum. Interestingly, genes involved in protein processing in the endoplasmic reticulum and antigen processing and presentation pathways are displayed in significance (*p* < 0.3, Table 1), supporting ERAP2N’s functional role of peptide processing for antigen presentation on HLA class I molecules to potentially modulate immune responses. This is the first reporting of all the genes that are affected by the expression level of ERAP2N.

The top 50 differential gene expression between ERAP2N-expressing TCs and cells that do not express ERAP2N is clearly represented in the heat map. There is a complete expression level reversal of these groups. Observing that cell-survival genes are up-regulated in TCs that do not express ERAP2N while simultaneously being down-regulated in ERAP2N-expressing TCs supports the hypothesis that ERAP2N-expressing TCs are highly prone to cell death prior to immune induced apoptosis, as previously reported [16]. Within these pathways, we have focused on genes that are associated with cell survival when expressed (ENO-3, CDC42, THBS3, SEC31B, LDHA, EGLN2, HLA-C, ITGB5, WNT7A, FN1) and with cell death (PPP1R12B, EGLN2). Interestingly, the current literature determined the function of these genes in the context of cancer and metabolism, which we could speculate given the similarity between cancer and placenta development [19,20,21,22,23,24,25,26,27,28,29,30,31,32,33,34]. As shown in Figure 5, the number of genes modulating cell survival are significantly increased compared to the genes promoting cell death. However, most of these homeostatically pro-survival genes are down-regulated in ERAP2N-positive TCs, thereby favoring cell death [19,20,21,22,23,24,25,26,27,28,29,30,31,32,33,34]. Even though SEC31B is named apoptosis-linked gene, the gene regulation seems to be complicated. When it is up-regulated, it promotes survival, but when it is down-regulated, it promotes cell death via apoptosis [34]. Interestingly, SEC31B was up-regulated with ERAP2N-expressing TCs. However, our balance diagram demonstrates nicely that the down-regulation of cell survival genes which promote cell death is more significant than the up-regulation of cell death genes. Our RNAseq analysis supports our in vitro data that cell death outcomes were significantly higher with ERAP2N-expressing TCs compared to those that did not express ERAP2N. Among these genes, we focused on human WNT7A because its coordinated expression is essential for the female reproductive tract development [3,36]. To date, most of the available data on WNTs’ gene expression in the uterine tissues concern rodents. In mice, Wnt7a is necessary for the appropriate expansion of endometrial glands and the organization of myometrium, as well as for the establishment of the gene expression of further WNT family members (Wnt4, Wnt5a) [3,36]. It has been proposed that WNT7A plays an important role in regulating uterine smooth muscle patterning and maintaining adult uterine function in mice [18]. In humans, high protein expression levels of WNT7A were observed in the cytoplasm and basal plasma membrane of the syncytium, indicating that WNT7A may be produced in syncytiotrophoblast cells and released toward fetal circulation [37]. The same study proposed that WNT7A might be expressed by villous stromal cells, possibly placental macrophages. Our results imply that ERAP2N expression may down-regulate WNT7A-mediated developmental processes, lowering the appropriate expansion of endometrial glands and the organization of the myometrium [18,37]. This translates into trouble during pregnancy. Figure 6 clearly demonstrates the association of ITGB5 and the WNT pathway, where, in ERAP2N-expressing TCs, both are down-regulated, which would promote cell death [15].

Lastly, PDIA3 was listed in the following KEGG pathways: protein processing in the endoplasmic reticulum and antigen processing and presentation pathways. Even though it was not in the top 50 DEGs and was not part of the top 12 genes we have focused on, it must be noted that PDIA3 is down-regulated in ERAP2N-expressing TCs. It was reported that PDIA3 expression was decreased in the TCs from women with PE, and decreased PDIA3 expression induced trophoblast apoptosis and repressed trophoblast proliferation by regulating the MDM2/p53/p21 pathway [38].

A direction for future study is to isolate the chemokines and cytokine signaling responsible for the apoptotic process observed due to ERAP2N expression, as it may help to direct future therapeutic targets and treatments. Running studies to identify the key immunologic cell types involved at each stage of apoptosis may similarly aid in illustrating the mechanism of immune targeting and eventual cell death in ERAP2N-positive TCs. Lastly, genetic screening for the ERAP2N of TCs may determine the success of the pregnancy, and a further evaluation of the genes involved in cell survival can help develop a therapeutic treatment of PE and other pregnancy-related disorders.

Collectively, this study provides further evidence for exploring mechanisms that could expand our knowledge of TCs in cancer and pregnancy, especially ERAP2-associated PE.

## 4. Materials and Methods

### 4.1. Cell Lines and pcDNA Stable Transfection of JEG-3 Cells

The JEG-3 trophoblast cell line, derived from gestational choriocarcinoma, was obtained from ATCC (Manassas, Virginia) and grown in T75 flasks with a MEM medium supplemented with 10% FBS and 1% Pen/Strep antibiotics at 37 °C, 5% CO2 for 24–48 h. For a stable transfection with mammalian expression empty vector pcDNA, 3.1 or pcDNA-ERAP2N, JEG-3 cells were plated at 3.0 × 10^4^ cells per well with appropriate media. They were allowed to attach overnight at 37 °C before transfection. The pcDNA plasmid with and without an ERAP2N insert was added at a concentration of 200 ng per well using Promega FuGENE^®^ Transfection Reagent (Promega #E2311). Untransfected cells were treated with FuGENE^®^ Reagent alone as a control. After 48 h, the cells were washed once, and the media changed to RPMI-1640 supplemented with 10% FBS(hi). Zeocin at 1 mg/mL concentration was added as a selection agent, and only the cells that survived were propagated. The ERAP2 protein expression level was confirmed by Western blot analysis as previously described [16].

### 4.2. RNA Sample Collection and Preparation

RNA was isolated from a cellular suspension of ERAP2N-positive trophoblast cells or ERAP2N-negative trophoblast cells. The suspended cells were rinsed with a PBS buffer. TRIzol reagent was added (1 mL per 5 × 10^6^ cells). The cell suspension was aspirated using a syringe to break up any noticeable clumps until the suspension became clear. Before use, samples were stored at −80 °C.

### 4.3. Clustering and RNA Sequencing

According to the manufacturer’s instructions, the clustering of the index-coded samples was performed on a cBot Cluster Generation System using a PE Cluster Kit cBot-HS (Illumina). After cluster generation, the library preparations were sequenced on an Illumina platform, and paired-end reads were generated. One microgram of RNA was used for the cDNA library construction at Novogene (Sacramento, CA, USA) using an NEBNext^®^ Ultra II RNA Library Prep Kit for Illumina^®^ (cat NEB #E7775, New England Biolabs, Ipswich, MA, USA) according to the manufacturer’s protocol. Briefly, mRNA was enriched using oligo(dT) beads followed by two rounds of purification and fragmented randomly by adding a fragmentation buffer. The first-strand cDNA was synthesized using a random hexamers primer. A custom second-strand synthesis buffer (Illumina), dNTPs, RNase H and DNA polymerase I were added to generate the second strand (ds cDNA). After a series of terminal repairs, poly-adenylation, and sequencing adaptor ligation, the double-stranded cDNA library was completed following the size selection and PCR enrichment. The resulting 250–350 bp insert libraries were quantified using a Qubit 2.0 fluorometer (Thermo Fisher Scientific, Waltham, MA, USA) and quantitative PCR. The size distribution was analyzed using an Agilent 2100 Bioanalyzer (Agilent Technologies, Santa Clara, CA, USA). The qualified libraries were sequenced on an Illumina NovaSeq 6000 Platform (Illumina, San Diego, CA, USA) using a paired-end 150 run (2 × 150 bases). Twenty million paired raw reads were generated from each library.

Paired-end reads were aligned to the hg38/GRCh38 reference genome using the Spliced Transcripts Alignment to a Reference (STAR) aligner v2.6.1 [39]. HTSeq v0.6.1 was used to count the read numbers mapped from each gene. RNA-seq counts were preprocessed and analyzed for differential expression using the edgeR v.3.30.0 [17], R package. *p*-values for differentially expressed genes were corrected using a False Discovery Rate (FDR) multiple testing correction method [40]. A functional enrichment analysis (KEGG) was performed using the enricher R package v.2.1 [41]. Row-median centered log2(TPM + 1) expression profiles for selected genes were visualized using the heatmap package v.1.0.12. All statistical calculations were performed within the R/Bioconductor environment v4.0.0 [42].

## 5. Conclusions

In conclusion, the expression of ERAP2N in trophoblast cells has a high potential to create a hostile uterine environment for fetal development and potentially result in an unsuccessful pregnancy. ERAP2N expression clearly reduces the expression of many cell survival genes and may increase the susceptibility to immune induced apoptosis via the up-regulation of genes involved in cell death, such as PPP1R12B and EGLN2. This is the starting point to define the mechanism of ERAP2N and how each of these significantly affected genes is involved in pregnancy success and failure. This knowledge will be important to develop genetic screening and therapeutic targeting to ensure healthy pregnancies for the mother and the baby.

## Figures and Tables

**Figure 1 ijms-22-08585-f001:**
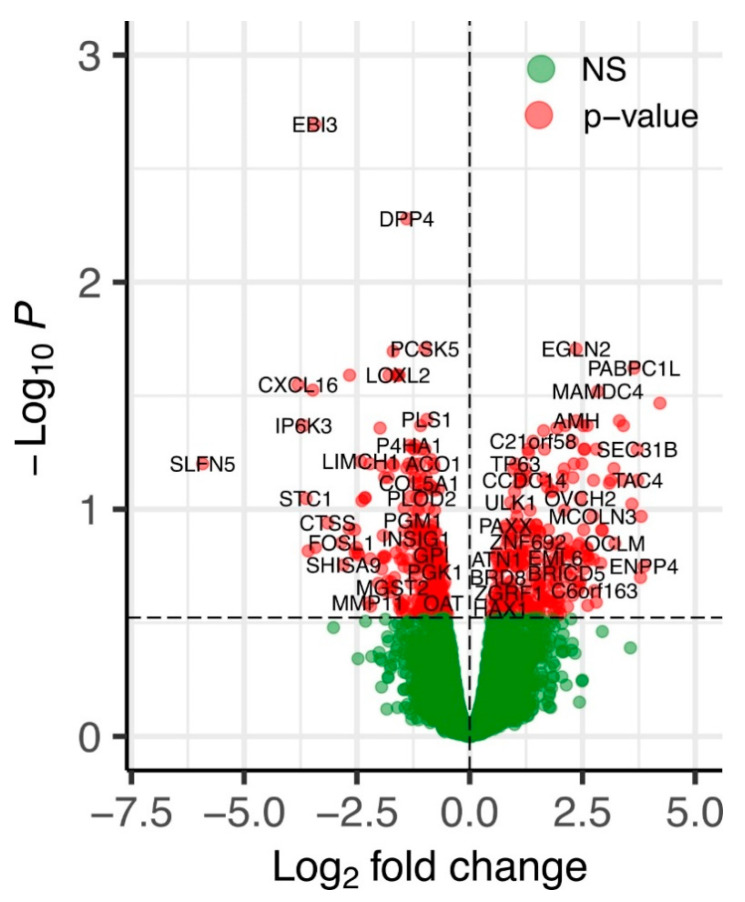
A volcano plot of differentially expressed genes in ERAP2N-positive vs. ERAP2N-negative trophoblast cells. X-axis—log2 fold change. Y-axis—(−log10) edgeR-calculated *p*-values. Red points correspond to significantly differentially expressed genes. Green points correspond to genes unchanged (NS: not significant) by ERAP2N expression level.

**Figure 2 ijms-22-08585-f002:**
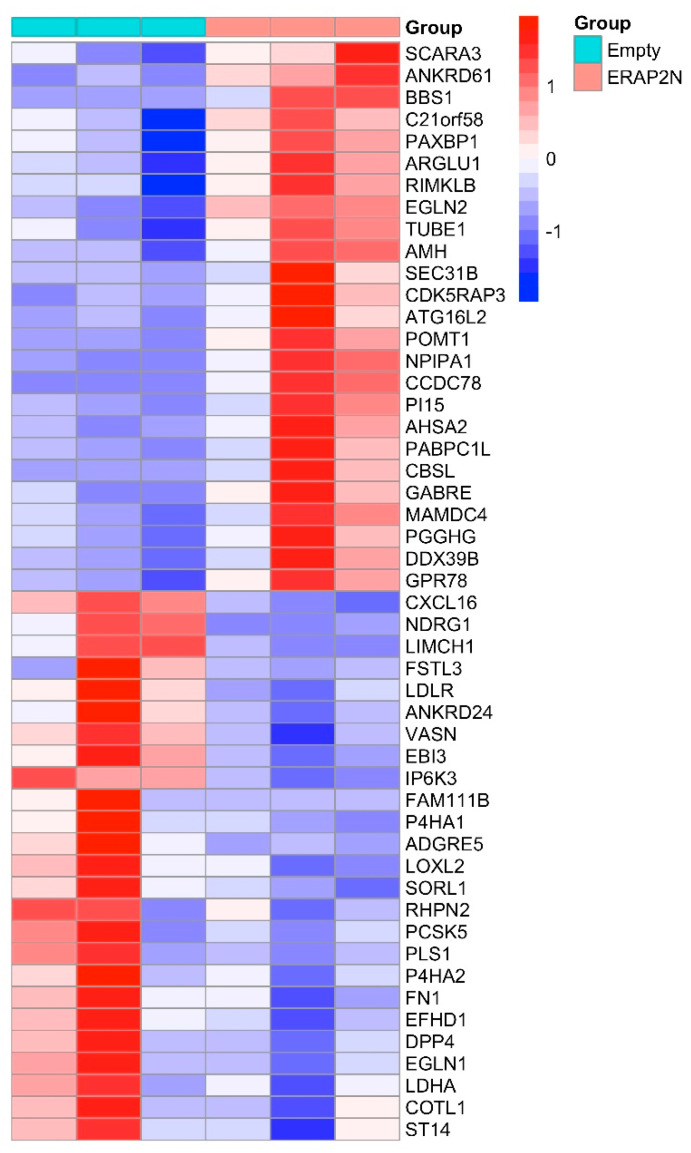
A heatmap of the top 50 most significantly differentially expressed genes (DEGs) in ERAP2N-positive vs. ERAP2N-negative trophoblast cells. The first three columns represent the cells transfected with an empty vector (no ERAP2N), and the last three columns represent ERAP2N-expressing trophoblast cells (TCs). Row-scaled log2-transformed TPMs (TPM = transcript count per million) are shown (N = 3/each group).

**Figure 3 ijms-22-08585-f003:**
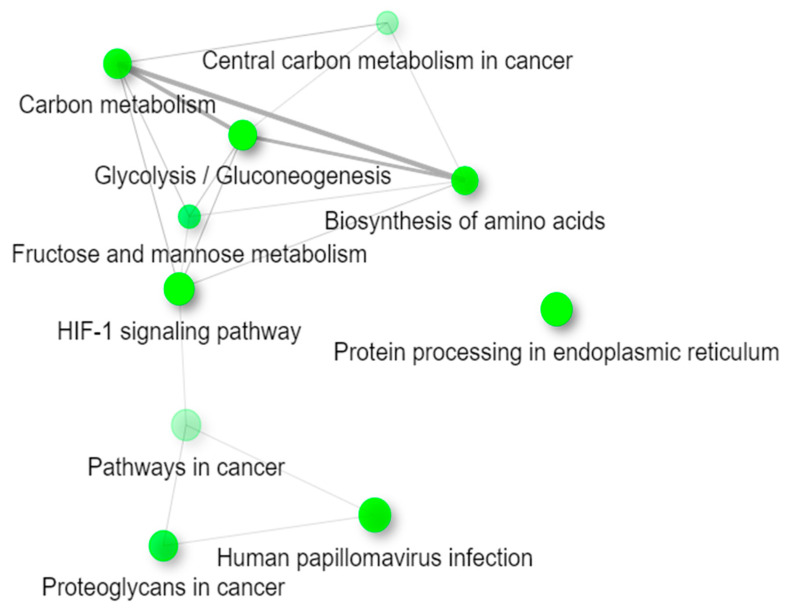
Web Diagram of the top 10 most significant KEGG pathways. This diagram represents how the differentially expressed genes listed in the enriched Kyoto Encyclopedia of Genes and Genomes (KEGG) pathways in Table 1 are connected. Two pathways (nodes) are connected if they share 20% or more genes. Darker nodes are more significantly enriched gene sets, and larger nodes represent larger gene sets. Thicker lines represent more overlapped genes between the pathways.

**Figure 4 ijms-22-08585-f004:**
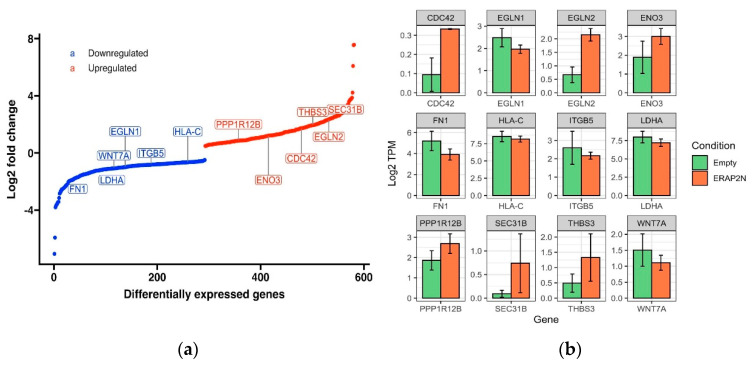
(**a**) Visualization of selected differentially expressed genes ranked by log2 fold change in ERAP2N-positive cells compared to ERAP2N-negative cells. The S-curve is a plot of genes differentially expressed in ERAP2N-positive cells compared to ERAP2N-negative cells, ranked by log2 fold change. The blue curve represents genes that are significantly down-regulated in ERAP2N-positive trophoblasts, while the red represents significantly up-regulated genes. (**b**) Bar Graph of Expression Level of Specific Genes from KEGG Pathways. Barplot of 12 differentially expressed genes, comparing the level of expression in ERAP2N-positive trophoblast cells to gene expression in ERAP2N-negative cells. Expression level reported as Log2 TPM (TPM = transcript count per million).

**Figure 5 ijms-22-08585-f005:**
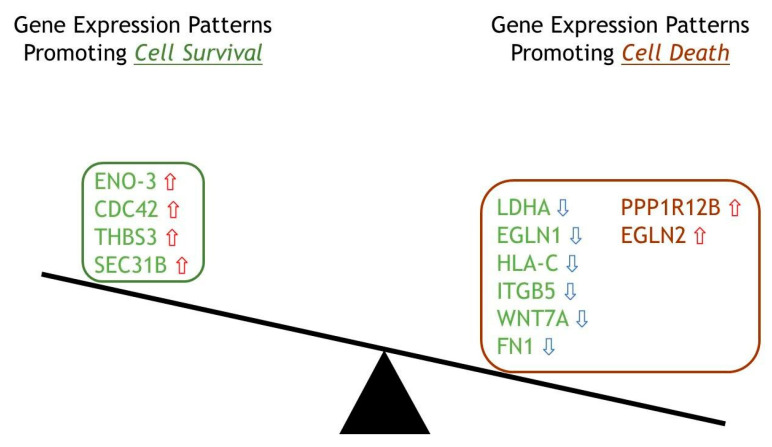
A balance diagram depicting significant differentially expressed genes classified into their role in the ERAP2N-positive trophoblast cell fate. Balance diagram of 12 significant differentially expressed genes divided into favoring either a cell survival or cell death role in the setting of ERAP2N. Green represents expression patterns favoring cell survival, and maroon represents expression patterns favoring cell death. Red arrows indicate that the gene was up-regulated, and blue arrows indicate that the gene was down-regulated in the setting of ERAP2N.

**Figure 6 ijms-22-08585-f006:**
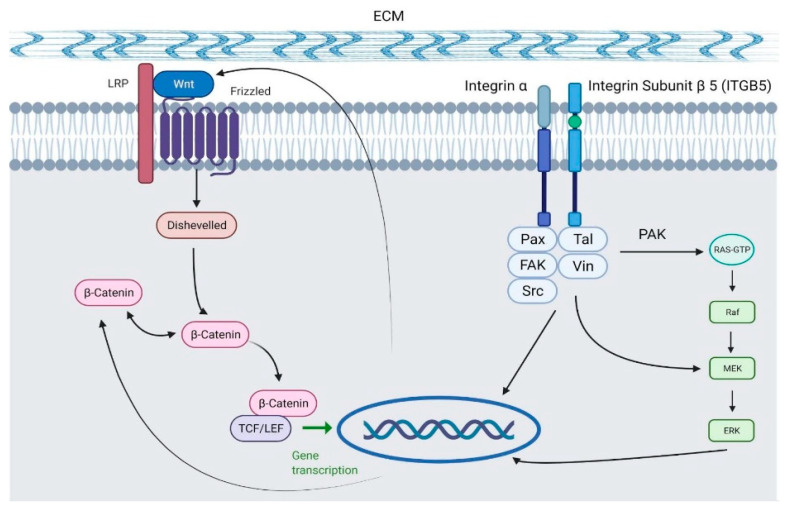
Diagram displaying the role of both ITGB5 and WNT in the transcription of genes to promote proliferation and differentiation [15] (Created with BioRender.com).

**Table 1 ijms-22-08585-t001:** Kyoto Encyclopedia of Genes and Genomes (KEGG) canonical pathways enriched in Differentially Expressed Genes (DEGs) with altered expression in the setting of ERAP2N. The underlined red (up-regulated) and blue (down-regulated) genes are the top 50 differentially expressed genes, appearing in multiple enriched pathways and having known functions that could potentially affect the trophoblast cell’s survival or death. The pathways also include genes that are not in the top 50 DEGs (genes in black).

Pathways	*p*-Value	Adjusted *p*-Value	Genes
Glycolysis/Gluconeogenesis	6.66 × 10^−8^	1.75 × 10^−5^	GPI; TPI1; PGAM1; ENO1; ***ENO3***; HK2; ***LDHA***; PKM; PGK1; ALDOC; ALDOA; PGM1; PFKP
HIF-1 signaling pathway	3.30 × 10^−5^	3.27 × 10^−3^	***EGLN1***; ***LDHA***; EGLN3; ***EGLN2***; TFRC; STAT3; PGK1; SLC2A1; ENO1; ALDOA; ***ENO3***; HK2
Fructose and mannose metabolism	3.73 × 10^−5^	3.27 × 10^−3^	PFKFB4; TPI1; AKR1B1; ALDOC; ALDOA; HK2; PFKP
Protein processing in the endoplasmic reticulum	9.74 × 10^−5^	6.41 × 10^−3^	ERO1A; PDIA3; HSPA5; WFS1; RRBP1; CKAP4; DDOST; PDIA4; HSP90B1; OS9; CALR; P4HB; SEC24D; ***SEC31B***; HSPA1B
Antigen processing and presentation	3.85 × 10^−4^	1.45 × 10^−2^	PDIA3; HSPA5; RFX5; ***HLA-C***; CALR; CTSS; HSPA1B; CTSB; TAPBP
Central carbon metabolism in cancer	5.68 × 10^−4^	1.66 × 10^−2^	***LDHA***; PKM; PGAM1; IDH1; SLC2A1; SLC16A3; HK2; PFKP
Proteoglycans in cancer	6.05 × 10^−3^	1.23 × 10^−1^	TGFB1; ***ITGB5***; FZD7; STAT3; ***FN1***; ***WNT7A***; ITPR2; ***CDC42***; WNT11; SDC1; ITGA5; EZR; ***PPP1R12B***
ECM-receptor interaction	9.79 × 10^−3^	1.84 × 10^−1^	***ITGB5***; ***FN1***; SDC1; COL9A3; ITGB6; ITGA5; ***THBS3***
Human papillomavirus infection	1.68 × 10^−2^	2.45 × 10^−1^	***ITGB5***; FZD7; ***FN1***; ***HLA-C***; ***WNT7A***; ***THBS3***; ***CDC42***; SLC9A3R1; PKM; WNT11; HEY1; ATP6V0A4; COL9A3; MAML3; ITGB6; ITGA5; TLR

## Data Availability

Not applicable.

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
