# Peer review of "Overexpression of ERAP2N in Human Trophoblast Cells Promotes Cell Death"

_ijms, 2021, doi:10.3390/ijms22168585_

Round 1

Reviewer 1 Report

To authors,

The paper is well written. The study is well designed and the interpretation is reasonable. I believe that this study would contribute a lot to better understanding of the placentology; placental physiology in both normal and pathological conditions. I have minor concern, of which clarification might improve the paper quality.

  1. Introduction should, as you may know, consist of three steps; known, unknown, and the problem (current hypothesis that should be solved/tested). You deployed the statement as such; however, “hypothesis” should be clearer. My concern is the relationship between the present aim (hypothesis) and the pathological condition (especially PE). You state something regarding PE in known and unknown section but not in hypothesis. As a matter of fact, you here did not “study” on PE but actually touched PE. 1) please state definitely whether you targeted PE here OR 2) you only touched PE in order to emphasize the “importance” of the present data/study-model (the tile of this special issue is “Placental Related Disorders of Pregnancy”). I never ask you that you should forcefully make mention of PE in the hypothesis. However, readers may be confused; 1) whether you here studied (at least a part of) PE, OR, 2) you simply “suggested” the application of the present data to the future study on PE. Please shortly describe your intention (present aim/hypothesis in relation with PE). You need not write long. Actually, you touched PE in the concluding remarks.
  2. You only used culture cells, right? Please state whether culture cells only (not having used human trophoblast cells (directly taken from the terminated specimens)) is sufficient to conclude things. Naturally, culture cells and “actual” human trophoblasts do not “behave” in an identical manner. Please shortly state the rationale of the present data regarding this point of view. You need not write long.

Author Response

Response to Reviewer 1 in RED:

Thank you for your suggestions and we have edited the manuscript according to your suggestions. 

To authors,

The paper is well written. The study is well designed and the interpretation is reasonable. I believe that this study would contribute a lot to better understanding of the placentology; placental physiology in both normal and pathological conditions. I have minor concern, of which clarification might improve the paper quality.

  1. Introduction should, as you may know, consist of three steps; known, unknown, and the problem (current hypothesis that should be solved/tested). You deployed the statement as such; however, “hypothesis” should be clearer. My concern is the relationship between the present aim (hypothesis) and the pathological condition (especially PE). You state something regarding PE in known and unknown section but not in hypothesis. As a matter of fact, you here did not “study” on PE but actually touched PE. please state definitely whether you targeted PE here OR 2) you only touched PE in order to emphasize the “importance” of the present data/study-model (the tile of this special issue is “Placental Related Disorders of Pregnancy”). I never ask you that you should forcefully make mention of PE in the hypothesis. However, readers may be confused; 1) whether you here studied (at least a part of) PE, OR, 2) you simply “suggested” the application of the present data to the future study on PE. Please shortly describe your intention (present aim/hypothesis in relation with PE). You need not write long. Actually, you touched PE in the concluding remarks.
    • Per your request, we have clarified that this is not a direct study of PE but rather it could give us important insight into understanding the pathophysiology of PE.
  2. You only used culture cells, right? Please state whether culture cells only (not having used human trophoblast cells (directly taken from the terminated specimens)) is sufficient to conclude things. Naturally, culture cells and “actual” human trophoblasts do not “behave” in an identical manner. Please shortly state the rationale of the present data regarding this point of view. You need not write long.
    • We also included in the introduction the justification for choosing a cell line study instead of the human specimen. See below.
    • "While the ultimate aim of our research is to apply these findings to live human subjects, culture cells were utilized in this experiment rather than human trophoblast cells from terminated specimens. This was due to the logistical and ethical difficulties associated with obtaining terminated specimens and logistical difficulty of obtaining Chilean population that would express ERAP2N, as well as that our experiment is a preliminary attempt to explore the differences, thus cultured cells are adequate for reaching our immediate goals. It is our hope that, in the future, the application of similar experimental protocols may be performed with human tissue to ensure the results are compatible with our initial reports in this study." 

Reviewer 2 Report

This manuscript is a nice study by Lospinoso et al, where the authors attempted to identify the cause of trophoblast cell death under ERAP2N over expression. I like the simple study design, yet an impactful outcome of this study and believe this would be important for the community. Hence, this could be a nice addition to MDPI-IJMS. However, I have few comments at this stage which are summarized below. If the other reviews are favorable, I would like the authors to discuss this in the discussion section before this can be published.

Comments:

(1) I think a major concern about this study is the mechanistic conclusion about what is causing cell death in trophoblasts. The authors claim at the beginning that ERAP2N overexpressing cells are prone to be killed by both cytotoxic T lymphocytes (CTLs) and Natural Killer cells (NKCs). However, at the end they just show that few pro-survival and anti-survival genes are differentially expressed in the ERAP2N overexpressing cells. How are these genes connected with cytotoxic immune-cells? There are no clear data that supports authors’ claim, or else, the authors need to do a better job while connecting these two aspects of their findings.

(2) The authors should also demonstrate the effect of ERAP2N downregulation in trophoblast cells. This is particularly important for the future therapeutic benefits for unhealthy pregnancies by modulating the levels of this gene in relevant cell types.

(3) Relating to comment 2 above, the authors need to describe and comment about the future impacts of the study with further details. They should highlight few key studies in this field and impress on the impact of their results in a point-by-point format.     

Author Response

Response to Reviewer 2 in RED:

We thank you for your suggestions. We have addressed each of the suggestions as follows:

This manuscript is a nice study by Lospinoso et al, where the authors attempted to identify the cause of trophoblast cell death under ERAP2N over expression. I like the simple study design, yet an impactful outcome of this study and believe this would be important for the community. Hence, this could be a nice addition to MDPI-IJMS. However, I have few comments at this stage which are summarized below. If the other reviews are favorable, I would like the authors to discuss this in the discussion section before this can be published.

Comments:

(1) I think a major concern about this study is the mechanistic conclusion about what is causing cell death in trophoblasts. The authors claim at the beginning that ERAP2N overexpressing cells are prone to be killed by both cytotoxic T lymphocytes (CTLs) and Natural Killer cells (NKCs). However, at the end they just show that few pro-survival and anti-survival genes are differentially expressed in the ERAP2N overexpressing cells. How are these genes connected with cytotoxic immune-cells? There is no clear data that supports the authors' claim, or else, the authors need to do a better job while connecting these two aspects of their findings. 

    • We have added the following statement in the manuscript to clarify the motivation and the justification of the study and how our study showed that ERAP2N weakens the TCs even before the immune induced cytotoxicity. “Even though our initial motivation to do this study was the observation of increased immune induced apoptosis with ERAP2N expression, the differential gene expression clearly indicated that ERAP2N expressing TCs are more prone to cell death due to decreased survival genes rather than ERAP2N improving immune targeting.”

(2) The authors should also demonstrate the effect of ERAP2N downregulation in trophoblast cells. This is particularly important for the future therapeutic benefits for unhealthy pregnancies by modulating the levels of this gene in relevant cell types. 

    • The empty vector TCs are the downregulation of ERAP2N control since normal TCs do not express ERAP2N. Thus, genetic screening for ERAP2N of TCs may determine the success of the pregnancy and further evaluation of the genes involved in cell survival can help develop therapeutic treatment to PE or other pregnancy related disorders.

(3) Relating to comment 2 above, the authors need to describe and comment about the future impacts of the study with further details. They should highlight few key studies in this field and impress on the impact of their results in a point-by-point format. 

    • We have highlighted WNT7A and PDIA3 and their studies to emphasize the importance of their roles in TCs and how they may relate to pregnancy success. We speculate that these can be used to genetically screen for early detection and potential therapeutic treatment in the future.
    • We have also added the below paragraph to our discussion to further highlight future directions:
    • "Directions for future study include replication of our experiment utilizing trophoblast tissue from  human samples rather than cell culture, to ensure reproducibility of the data. Additionally, isolating the chemokines and specific chemical signaling responsible for the apoptotic process observed due to ERAP2N expression should be pursued as it may help to direct future therapeutic targets and treatments. Running studies to identify the key immunologic cell types involved at each stage of apoptosis may similarly aid in illustrating the mechanism of immune targeting and eventual cell death in ERAP2N positive TCs."

Reviewer 3 Report

In the current paper, the authors present their results regarding an isoform of ERAP2-N gene expressed in trophoblast cells. The authors suggest that the ERAP2-N expression in trophoblast cells create a “hostile uterine environment for fetal development” and may explain at least in part an unsuccessful pregnancy.

Since the manuscript brings new knowledge in this specific field, I consider that it can be accepted after minor text revision.

Minor comments:

Please finish the following phrases:

2.2 An overview of differentially expressed genes … in…

Figure 1. A volcano plot of differentially expressed genes … in…

2.3 The heatmap of the top 50 genes … in…

Figure 2. A heatmap of top 50 most significantly differentially expressed genes … in…

2.5 Differentially expressed genes (DEGs) … in…

Figure 4. (a) Visualization of selected differentially expressed genes … in…

2.6 Top 12 genes involved in cellular fate and expressed in …

For example:  “in human TCs that overexpressed ERAP2N as compared to the TCs that do not express ERAP2N” …etc

Author Response

Response to Reviewer 3 in RED:

In the current paper, the authors present their results regarding an isoform of ERAP2-N gene expressed in trophoblast cells. The authors suggest that the ERAP2-N expression in trophoblast cells create a “hostile uterine environment for fetal development” and may explain at least in part an unsuccessful pregnancy.

Since the manuscript brings new knowledge in this specific field, I consider that it can be accepted after minor text revision.

Minor comments:

Please finish the following phrases:

2.2 An overview of differentially expressed genes … in…

Figure 1. A volcano plot of differentially expressed genes … in…

2.3 The heatmap of the top 50 genes … in…

Figure 2. A heatmap of top 50 most significantly differentially expressed genes … in…

2.5 Differentially expressed genes (DEGs) … in…

Figure 4. (a) Visualization of selected differentially expressed genes … in…

2.6 Top 12 genes involved in cellular fate and expressed in …

For example:  “in human TCs that overexpressed ERAP2N as compared to the TCs that do not express ERAP2N” …etc

    • We thank you for your suggestions. We have edited all of the ones you have suggested to keep them consistent throughout the manuscript.